# Cases of Hemorrhagic Fever with Renal Syndrome in Russia during 2000–2022

**DOI:** 10.3390/v15071537

**Published:** 2023-07-12

**Authors:** Evgeniy Tkachenko, Svetlana Kurashova, Alexandra Balkina, Alexander Ivanov, Mariya Egorova, Oksana Leonovich, Yulia Popova, Rostislav Teodorovich, Alla Belyakova, Petr Tkachenko, Dmitriy Trankvilevsky, Ekaterina Blinova, Aydar Ishmukhametov, Tamara Dzagurova

**Affiliations:** 1FSASI “Chumakov Federal Scientific Center for Research and Development of Immune-and-Biological Products of Russian Academy of Sciences” (Institute of Poliomyelitis), 108819 Moscow, Russiaale-bal@yandex.ru (A.B.); dzaguron@gmail.com (T.D.); 2Department of Internal Disease Propaedeutics, I.M. Sechenov First Moscow State Medical University, 119991 Moscow, Russia; 3Federal Center of Hygiene and Epidemiology, 117105 Moscow, Russia; 4Department of Genetic Engineering and Biotechnology, Central Research Institute of Epidemiology, 111123 Moscow, Russia

**Keywords:** orthohantavirus, hemorrhagic fever with renal syndrome, zoonotic viral diseases, epidemics, morbidity, incidence rate, case-fatality rate

## Abstract

During 2000–2022, a total of 69 of Russia’s 85 administrative regions reported 164,580 hemorrhagic fever with renal syndrome (HFRS) cases, with an annual average rate of 4.9 cases/100,000 population (10^5^ popul.). European Russia reported 162,045 (98.5%) cases in 53/60 regions with 9.7 cases/10^5^ popul. Asian Russia reported 2535 (1.5%) cases in 16/25 regions with 0.6 cases/10^5^ popul. In the same period, Russia reported 668 (0.4%) fatal HFRS cases, and 4030 (2.4%) cases among children under the age of 14 years. Most HFRS cases occurred during autumn and winter. The incidence among rural residents was 6.7 per 10^5^ popul., higher than the urban 4.4 per 10^5^ popul.; however, among HFRS patients, rural and urban residents account for 35% and 65%, respectively. Six hantaviruses, causing HFRS of different clinical severity, were recognized as pathogens: Hantaan (HTNV) and Amur (AMUV) of *Orthohantavirus hantanense* species, Seoul (SEOV) of *Orthohantavirus seoulense* species, Puumala (PUUV) of *Orthohantavirus puumalaense* species, and Kurkino (KURV) and Sochi (SOCV) of *Orthohantavirus dobravaense* species, with the principal hosts *Apodemus agrarius coreae*, *Apodemus peninsulae*, *Rattus norvegicus*, *Myodes glareolus*, *Apodemus agrarius agrarius*, and *Sylvaemus ponticus*, respectively. It was found that 97.7% of HFRS cases are caused by PUUV, therefore, this virus plays the main role in the HFRS morbidity structure in Russia.

## 1. Introduction

The clinical features of the first hemorrhagic fever with renal syndrome (HFRS) cases, called “hemorrhagic nephroso-nephritis”, in Russia were described in 1935 [1,2]. It was believed that the area of the disease distribution was limited to the Far-Eastern part (the Amur River basin) of Russia. Therefore, retrospectively, the so-called “Tula fever” known since 1930, can be considered as the first HFRS discovery in Russia [3,4]. 

This disease was been detected in different regions of European Russia during the 1950s. Fever cases were recorded and further classified using geographical terms (Tula, Yaroslavl, Ural fevers, etc.) [5,6]. At the same time, in Russia, instead of “Hemorrhagic nephroso-nephritis”, a new term for the disease “Hemorrhagic fever with renal syndrome” was proposed [7]. 

Despite establishing the viral etiology of HFRS in experiments on reproducing the HFRS clinical features in human [8,9,10], numerous attempts by Russian and foreign researchers to isolate and cultivate the HFRS pathogen in in vitro remained unsuccessful for a long time. After more than 40 years of research, in the late 1970s, the HFRS virus was first isolated by H.W. Lee, P.W. Lee, and K. Johnson in South Korea from the field mouse (*Apodemus agrarius coreae*) [11].

Since the isolation of the first orthohantavirus, at least 40 other orthohantaviruses have been discovered [12]. In humans, orthohantaviruses cause two diseases: HFRS in Eurasia, and hantavirus pulmonary syndrome in the Americas [13,14]. 

HFRS has the highest incidence rate among reportable zoonotic viral diseases in Russia. HFRS has been included in the official reporting system of the Russian Ministry of Public Health in 1978. There were a total of 292,353 registered HFRS cases in Russia from 1978 to 2022 [15].

The number of reported hantaviral infection cases are increasing in many countries and new orthohantavirus strains have been successfully identified worldwide [16,17,18,19]. In addition, people’s lifestyle mobility exacerbates the infection risks. That is why hantaviral infections might be underestimated, even in countries where the disease is known, as it has an asymptomatic and non-specific mild clinical course, and there is a lack of simple standardized laboratory diagnostics in hospitals [20]. Therefore, hantaviral infections are a global public health problem.

This review summarizes the recent advances in the etiology, epidemiology, epizootiology, laboratory diagnostics, control and prevention of HFRS in Russia.

## 2. HFRS Incidence Rate and Geographic Distribution

During 2000–2022, a total of 69 of Russia’s 85 administrative regions reported 164,580 HFRS cases, with an annual average rate of 4.9 cases per 10^5^ popul. [15] (Table 1). 

HFRS cases were distributed unevenly throughout the country. However, different geographical regions are distinguished by the morbidity rates of HFRS which is mainly associated with the landscape zones (mixed and broad leaved forests, forest-steppe, steppe, semi-desert, desert and mountainous), rodent species-reservoir of orthohantaviruses, feed products, etc. [21]. 

European Russia reported 162,045 cases (98.5% of all morbidity in Russia) in 53 out of 60 regions, with an annual average rate of 9.7 cases per 10^5^ inhabitants (Table 1). The highest HFRS incidence rates occurred in the Ural and Middle Volga areas (Figure 1). 

In this area, there are 14 administrative regions with high HFRS morbidity (>10 per 10^5^ popul.), including Udmurtia and Bashkiria regions, with the annual average incidence rate of 57.6 and 44.5 cases per 10^5^ popul., respectively [15].

Overall, 139,174 (84.5%) of HFRS cases, including 480 (72%) case-fatality, and 3603 (89.4%) cases of infection in children under 14 years were reported from these 14 regions. 

Of note, in this area, there is an excellent fodder potential for the bank voles, the reservoir hosts of *Orthohantavirus puumalaense* species, Puumala (PUUV), due to the unique geo-botanical conditions, particularly the lime-forests.

A total of 16 out of 25 Asian administrative regions reported 2535 (1.5%) HFRS cases with an annual average incidence rate of 0.6 cases per 10^5^ popul. (Table 1). 

Overall, 2229 (88%) of HFRS cases were registered in six out of eight Far-Eastern administrative regions, including mainly four regions: Vladivostok (1257 cases, 2.7 per 10^5^ popul.), Khabarovsk (634 cases, 2.0 per 10^5^ popul.), Amursk (100 cases, 0.6 per 10^5^ popul.) and Jewish (236 cases, 5.8 per 10^5^ popul.) [15].

In Siberia, 300 HFRS cases were registered in 10 out of 17 administrative regions, with an annual average incidence rate of 0.4 per 10^5^ popul.; 6 out of 11 regions were in Western Siberia. The highest HFRS incidence rates occurred in Yugra (159 cases, 0.5 per 10^5^ popul.), followed by Yamalia (116 cases, 1.1 per 10^5^ popul.), and Tyumen (21 cases, 0.1 per 10^5^ popul.) [15]. It should be emphasized that in Yugra, Yamalia and Tyumen regions, most of Russia’s oil and gas industry is concentrated. Bogs comprise a considerable part of the landscape. Practically, there are no favorable conditions for dwelling of mammals. The retrospective analysis of some of HFRS cases in West Siberian regions has shown that most of them were imported. HFRS infected persons came to these areas as temporary oil and gas field workers from other endemic European regions (for example, neighboring Udmurtia and Bashkiria), where they probably were infected with PUUV [22]. 

HFRS morbidity analysis during the previous two decades showed a steady, non-decreasing trend in Russia (Figure 2). Human HFRS epidemics were characterized by one-year cycles with a frequency of 3–4 years (sometimes 2 years).

At the same time, two-year peaks due to asynchronous manifestation HFRS epidemic activity have been observed even in neighboring territories, for example, the Bashkiria and Udmurtia regions (Figure 3).

Though most of HFRS cases in Russia occurred during autumn and winter, the infection was persistent all throughout the year (Figure 4).

A significant decrease in feed products during autumn and winter seasons in the natural habitats of rodents-carriers of orthohantaviruses forces them to migrate to people’s homes, which leads to an increase in their contacts with humans and an increase in the incidence of HFRS [23].

In general, HFRS incidence rate per 10^5^ popul. in Russia is higher among rural residents (6.7 per 10^5^ popul.) as compared to urban (4.4 per 10^5^ popul.), however, among patients with HFRS, rural and urban residents account for 35% and 65%, respectively. There were a few exceptions to this general trend. The absolute HFRS cases number was higher among rural populations in five Central European Russian regions, where HFRS outbreaks were caused by the Kurkino virus (KURV) (*Orthohantavirus dobravaense* species), as well as in two Far-Eastern regions, Amur and Jewish, where HFRS morbidity was caused by the Hantaan virus (HTNV) (*Orthohantavirus hantanense* species) (57% and 58% of the overall cases found among rural population, respectively) [24,25].

In general, in Russia, HFRS appears to most frequently affect persons from 20 to 50 years old, more often males than females; children under the age of 14 years make up an average rate of 2.4% of the cases (4030 cases). The overall case-fatality rate was 0.4% (668 total cases), varying from 0.4% in the Central European Russia regions to 7–14% in the Black Sea coast area of European Russia and Far-Eastern regions. This is due to the Sochi virus (SOCV) (*Orthohantavirus dobravaense* species) and HTNV which are the main and the most pathogenic hantaviruses in these areas [21,22]. 

Serological examination results of convalescents who had the illness more than 25 years ago have shown a long-term hantavirus-specific antibodies preservation [23]. Orthohantavirus antibody prevalence rate in Russia was found to be different: in European Russia, the average rate was 5.5% (varying between regions from 0.1% to 12%) while it was 1.5% (0.1–3.7%) in Far-Eastern Russia. The highest seroprevalence was observed in regions with great intensity of HFRS natural foci (up to 30% in Bashkiria).

Among the randomly selected healthy population, the male/female ratio is 2:1, but among the HFRS patients, this ratio is 4:1 [24]. The difference may be explained primarily by the fact that in women HFRS is frequently misdiagnosed as pyelonephritis and other diseases [26]. Moreover, milder, even asymptomatic course of the infection in women cannot be excluded either. More frequent antibody findings in older subjects may be explained by the increasing number of human contacts with infection sources over lifetime, indicating the endemicity of the area. The orthohantavirus antibodies detection in sera of people without reported HFRS may be explained by milder, even asymptomatic clinical infection course as well as by the low awareness of local physicians.

The evidence of the mode of orthohantavirus transmission to humans is derived principally from epidemiological observations. Experimental findings concerning the orthohantavirus transmission between rodents are auxiliary data that shed light on the way the virus is transmitted to humans. The orthohantavirus infection pathogenesis in bank voles (*Myodes glareolus)* indicates that the virus is persistent in reservoir animals, causes chronic, apparently asymptomatic disease, in these hosts and this may be shed over a long period with urine, feces and respiratory secretions [27]. Aerosols are sufficient to transmit orthohantavirus horizontally among rodents. Evidence of orthohantavirus infection respiratory route was demonstrated during two laboratory outbreaks in Tula [6] and in Moscow [28] involving 12 and 114 HFRS cases, respectively. The source of unforeseen air-borne infections were large shipments of bank voles brought to the animal facilities from natural foci of infection and kept in large cages for 1–3 months. In a number of cases, the air-borne transmission could be the only possible explanation of human infection. Whether orthohantavirus infection is related to ingestion of food contaminated with the virus [29], or to contact with mucous membranes, or to contamination of breaks in the skin barrier have not been clearly evaluated. 

The numerous epidemiological studies of infections acquired in natural conditions suggest that close human contact with rodents should be a risk factor for infection by orthohantavirus. The risk groups are persons who are permanently working or frequently visiting HFRS foci. There is no evidence of interhuman, or secondary HFRS spread in Russia. Person-to-person orthohantavirus transmission as well as transmission through inadvertent self-inoculation by hospital staff handling viremic blood have not yet been demonstrated.

Until the late 1990s, HFRS cases in European Russia were thought to be associated with PUUV only. However, during the last two decades, HFRS cases caused by KURV were detected in seven regions of Central European Russia. There, three major HFRS outbreaks associated with KURV were reported. A total of 950 cases were registered, including 130 cases in 1991, 211 cases in 2001, and 609 cases in 2006. Detailed investigations of the outbreaks had revealed the western subtype of striped field mouse (*Apodemus agrarius*) as the virus reservoir and KURV as the causative infectious agent.

The results of comparative HFRS epidemiological data analyses caused by PUUV (HFRS-PUUV) and KURV (HFRS-KURV) outbreaks indicated that 97% of total HFRS-KURV cases were registered among the rural population and only 3% among the urban population. At the same time, 30% of HFRS-PUUV cases in Bashkiria were registered among the rural population and 70% among the urban population.

Most HFRS-PUUV cases were registered during the period from August to December with HFRS human epidemic peaking in October, while HFRS-KURV cases were registered during the period from November to March, with the peak in December. The clinical differences between HFRS-PUUV and HFRS-KURV were characterized by different frequency and severity of HFRS pathognomonic symptoms manifestation. In HFRS-PUUV area, the main risk factors of infection are short-time stay in the forest (55%), gardening, and farming activities (36%), while those in HFRS-KURV area are connected with hibernal cattle breeding (73%) and other agricultural activity (25%). It is necessary to mention that during the HFRS-KURV outbreak in 2006–2007 in three regions, both viruses, KURV (87.8% of all cases) and PUUV (12.2%), were recognized as the causative agents of the HFRS cases [21,30,31,32,33,34,35,36,37,38]. 

During 2000–2017, SOCV circulation, as well as HFRS patients infected by this virus, were found in the Sochi region, which is the southern part of Russia [35,39,40,41,42,43,44]. A new SOCV associated with the Black Sea field mouse (*Sylvaemus ponticus*) as a novel natural orthohantavirus host was recognized as the causative agent of the human infections. *Sylvaemus ponticus* is a naturally occurring genus in the Southern European Russia and Transcaucasian countries between the Black and the Caspian Sea. Cell culture isolates of SOCV have been generated from *Sylvaemus ponticus* and HFRS patients with fatal outcomes [34,41]. For the period from 2000 to 2017, 7 of Krasnodar region’s 37 districts reported 70 HFRS cases, including 38 HFRS cases in the Sochi district.

Based on the present state of knowledge, SOCV seems to be the most dangerous in comparison with the other three viruses (Dobrava (DOBV), KURV, and Saarema (SAAV)) of the *Orthohantavirus dobravaense* species. The case-fatality rate was determined to be as high as 14%. Nearly 60% of clinical courses were defined as severe (including deaths) and nearly 40% as moderate. Four times more males than females were affected. Though quite unusual for orthohantavirus disease, young people also became ill due to SOCV infection; 10% of the patients were found to be between 7 and 15 years old and the average age of all patients was not much higher than 30 years.

Overall, six orthohantaviruses causing HFRS were recognized as pathogens in Russia: HTNV, Amur virus (AMRV), Seoul virus (SEOV), PUUV, KURV and SOCV. HFRS cases in Asian Russia (mainly in Far-East) are caused by HTNV, AMRV and less often by SEOV. The principal hosts of these viruses are the eastern subtype of striped field mouse (*Apodemus agrarius*), Korean field mouse (*Apodemus peninsulae*), and Norway rat (*Rattus norvegicus)*. HFRS cases registered in European regions are caused mainly by PUUV associated with bank vole (*Myodes glareolus*) and to a less extent by KURV associated with the western subtype of striped field mouse *(Apodemus agrarius* agrarius) and SOCV associated with the Black Sea field mouse (*Sylvaemus ponticus*). 

The comparative analyses results of clinical HFRS caused by six orthohantaviruses, mentioned above, indicated that infection by all these viruses can cause three clinical forms of the disease: mild, moderate and severe. There are certain differences in the frequency of these forms for different orthohantaviruses. The severe forms detection rate as well as the case-fatality rates are significantly higher for SOCV (14%) and HTNV (5–8%) than for PUUV, SEOV, and KURV (up to 1%) [45].

Of note, 97.7% of HFRS cases registered in Russia were caused by PUUV [46], which explains the overall low case-fatality rate in the country. Only 2.3% of HFRS cases were caused by other five orthohantaviruses. Thus, PUUV plays the main role in the HFRS morbidity structure in Russia.

## 3. Epizootiology 

To identify the small mammals that can serve as orthohantavirus natural hosts and potential vectors, more than 130,000 animals belonging to 80 species trapped practically in all the landscape zones of the Russian territory were examined for the hantaviral antigen presence using the ELISA test [47]. 

The hantaviral antigen was detected in 43 species relating to six families (Talpidae, Soricidae, Sciuridae, Cricetidae, Muridae, and Gliridae) of two orders (rodentia and insectivora) including in 40 species in the European part and 29 species in the Asian part of Russia. In addition to small mammals, hantaviral antigen was detected in the lungs of 13 species of birds trapped in the Far-Eastern Russia: grey heron (*Ardea cinerea*), striated heron (*Butorides striata*)*,* Eurasian jay (*Garrulus glandarius*), black-faced bunting (*Emberiza spodocephala*), yellow-throated bunting (*Emberiza elegans*), coal tit (*Parus ater*), marsh tit (*Poecile palustris*), common pheasant (*Phasianus colchicus*), Daurian redstart (*Phoenicurus auroreus*), wood nuthatch (*Sitta europaea*), Ural owl (*Strix uralensis*), rufous turtle dove (*Streptopelia orientalis*), and hazel grouse (*Tetrastes bonasia*) [31,47,48,49,50,51,52,53,54,55].

Examination of the small mammals showed that practically each landscape zone has natural foci with different hantavirus activity degrees. The most active natural foci are situated in the central and eastern regions of European Russia, in a zone of broad-leafed forests and forest-steppe, and also at the different landscape borders (taiga forests, mixed and broad-leafed ones, and then forest-steppe). 

In active natural foci, practically all species of rodents and insectivorous (even rare ones) are involved in the infectious process, though to a much lesser extent than the main reservoir hosts, which are usually the dominant species of particular landscape formations [54].

The vast majority of rodents and insectivore species as well as bird species harboring orthohantaviruses are probably random, spill-over hosts. Alongside this, the epidemiological significance of certain rodents is established now in different Russian regions. HFRS cases in Far-Eastern Russia are mainly caused by HTNV, AMRV and less often by SEOV. The principal hosts of additional viruses (not shown to be pathogenic in humans), Tula virus (TULV), Khabarovsk virus (KHAV) and Topografov virus (TOPV), are the common vole (*Microtus arvalis*), reed vole (*Microtus fortis*), and Siberian lemming (*Lemmus sibiricus*), respectively.

Recently, a novel orthohantavirus has been discovered in the Black Sea coast area of European Russia [56]. The obtained data showed that Major’s pine vole (*Microtus majori*), is a newly recognized natural orthohantavirus host. The newfound virus, called Adler (ADLV), is related to TULV and has been classified as an *Orthohantavirus tulaense* species. Nevertheless, ADLV and TULV occurrence in the same region suggests that ADLV is not only a geographical variant of TULV but a host-specific taxon. High intracluster sequence variability indicates long-term presence of the virus in this region. The ADLV pathogenic potential needs to be determined.

## 4. Specific Laboratory Diagnostics of HFRS

A complex of virological, immunological and molecular genetic methods was used to detect orthohantaviruses and specific antibodies in HFRS patients and small mammals [35,39,45,57,58,59].

More than 1000 samples from 16 small mammal and 13 bird species, and 300 blood samples from HFRS patients were collected in the acute phase. About 80 organ samples from sectional organs of HFRS patients were examined for orthohantaviruses isolation in VERO-E6 cells.

As a result, 76 hantaviral strains were isolated, including 53 strains from five species of small mammals belonging to the Cricetidae family (bank vole, *Myodes glareolus*; red-backed vole, *Craseomys rufocanus*; ruddy vole, *Myodes rutilus;* common vole, *Microtus arvalis*; and reed vole, *Microtus fortis*) as well as four species of the Muridae family (striped field mouse, *Apodemus agrarius*; Korean field mouse, *Apodemus peninsulae*; Black Sea field mouse, *Sylvaemus ponticus*; and Norway rat, *Rattus norvegicus*). Also, 1 strain from the yellow-throated bunting bird, *Emberiza elegans*; 10 strains from the HFRS patients’ blood and 12 strains from HFRS patients’ sectional organs were isolated (Table 2) [25,31,40,50,52,58,59,60,61,62]. With the exception of the bronchi, orthohantavirus was detected in all the organs examined (lungs, liver, spleen, kidney, heart, brain, and lymph nodes), indicating a wide distribution of orthohantavirus in the HFRS patients’ organs.

Until the end of the twentieth century, the diagnosis of HFRS in Russia was based on clinical symptoms and, if the infection was fatal, on the pathoanatomical organs examinations of the deceased HFRS patients. Correct HFRS diagnosis depended on the qualification of the medical staff. HFRS is characterized by a wide range of clinical symptoms (more than 70) [4], which greatly complicates the differential HFRS diagnosis, often leading to (sometimes tragic) errors in clinical diagnosis.

To develop an optimal scheme for serological testing of HFRS-PUUV patients in European and HFRS-HNTV patients in the Far-East region of Russia, 6000 samples of blood sera from HFRS patients and convalescents from 27 Russian administrative regions and 2000 samples of blood sera from patients with other diagnoses but similar clinical picture were examined. 

It has been shown that antibodies from HFRS patients (both European and Far Eastern foci) can be detected via indirect IFA from 2–3 days after disease onset and they reach peak levels at the end of the second week. The tendency for a slow decrease in antibody titers was observed quite late (six months after the onset of the disease in the Far-East region and several years after in the European part). 

The dynamics of IgM and IgG antibody formation to PUUV and HNTV showed that antibodies of these classes were detected in the early days of illness, with IgM titers significantly exceeding IgG titers in the early period. In some patients, IgM was undetectable in paired sera as early as 45 days after disease onset, but there were patients in whom antibodies of this class persisted for a long time and, in individual cases, were detected at quite high titers a year after disease onset. Therefore, the diagnostic level of IgM detection can be discussed only on the basis of seroconversion dynamics in the study of paired blood sera and, in controversial cases, in the simultaneous study of IgG seroconversion [59,63].

Serological confirmation of the HFRS clinical diagnosis in Russia as a whole was 86.5%. Discrepancies between specific serological and preliminary clinical diagnoses were associated with errors in clinical diagnosis.

The use of a specific laboratory HFRS diagnosis made it possible to confirm the assumption of clinicians about the possible existence of mild and erased forms of the infection course. The presence of mild and erased HFRS forms, as well as the gross sinfulness of clinical diagnostics, determine the natural immune layer of the population, the value of which reflects the level of clinical and serological HFRS diagnosis and the number of unrecorded patients who have HFRS under different diagnoses.

## 5. Control and Prevention

Periodical and massive reproduction of rodents, forming epizootics among them, is the main and determinative factor that causes HFRS epidemics in humans. HFRS disease prevention mainly includes measures aimed at reducing exposure to rodents and their excreta. However, rodent control measures are expensive and difficult to maintain over long periods. It is practically impossible to eradicate the orthohantavirus-carrying rodent hosts from nature. 

Hence, the most prospective and effective measure for decreasing HFRS morbidity in endemic Russian regions could be regular and massive vaccine prophylaxis against the disease. HFRS morbidity may be used as an indicator to estimate the required HFRS vaccine doses for vaccination in Russia. In the 20 European regions (where HFRS is caused mainly by PUUV and less by KURV and SOCV), there is a population of about 45 million and in the 4 Far-Eastern regions (where HFRS is caused mainly by HTNV, AMRV and less by SEOV), there is a population of about millions; thus, these populations potentially need the vaccination against HFRS. 

Despite intensive research efforts, no licensed vaccine is currently available in Europe and America. The only widely distributed vaccines are the inactivated vaccines used in China and South Korea which are directed against HTNV and SEOV [64]. However, these vaccines probably do not efficiently protect against PUUV virus infection and therefore are not suitable for vaccination in European Russia.

At present, VERO cell-derived, Puumala–Hantaan virus-combined inactivated vaccine against HFRS is being tested in pre-clinical studies in Russia. Antibody response in vaccinated laboratory animals showed good neutralizing antibody production against the above-mentioned viruses [65].

## 6. Conclusions

HFRS represent a serious medical problem in Russia, especially in the European part of the country, where the total number of cases as well as the incidence rates are by far the highest in Europe. There are strong regional differences in HFRS incidence rates across the country, reflecting the differences in small mammals composition and consequently the differences between the causative agents. At least six causative agents of HFRS have been identified in Russia: Hantaan, Amur, Seoul, Puumala, Kurkino and Sochi. This unique combination of various causative agents together with the high number of cases leads to the accumulation of highly valuable epidemiological data and allows to understand the epidemiological and clinical differences between the various HFRS forms.

## Figures and Tables

**Figure 1 viruses-15-01537-f001:**
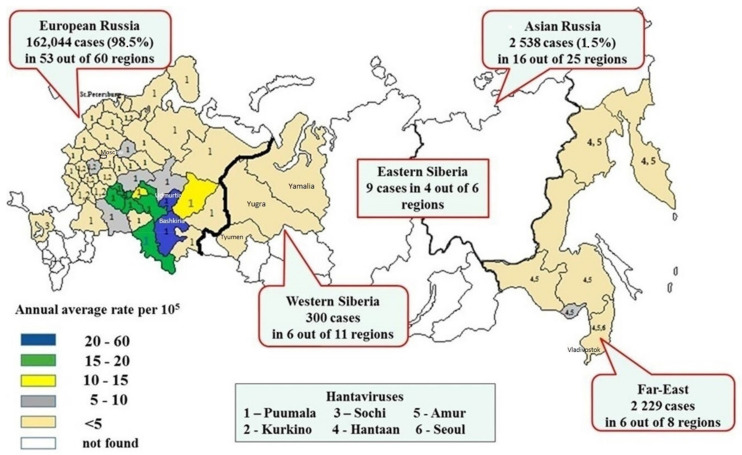
Distribution of HFRS morbidity in Russia (2000–2022).

**Figure 2 viruses-15-01537-f002:**
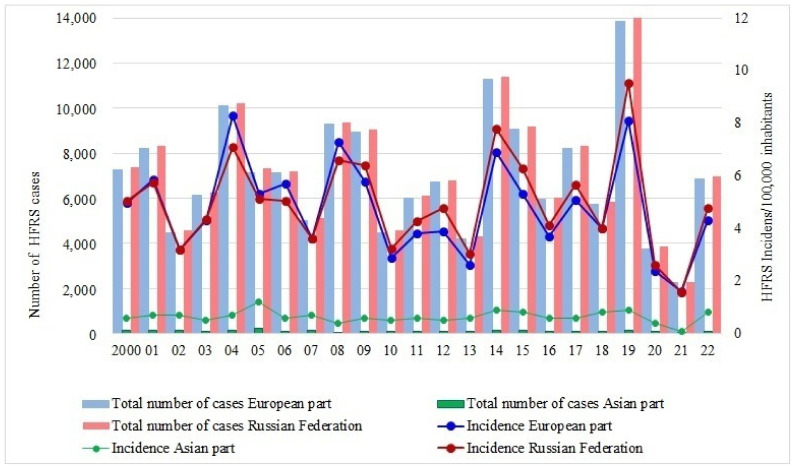
The dynamics of HFRS morbidity in Russia (2000–2022).

**Figure 3 viruses-15-01537-f003:**
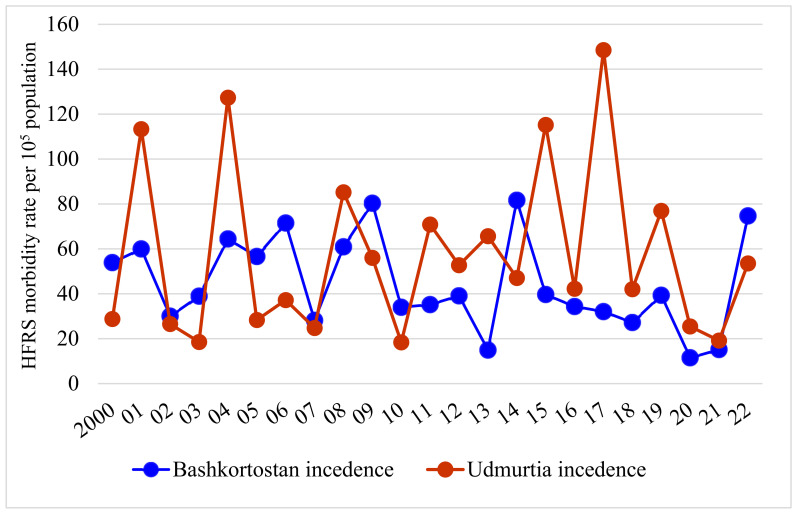
Asynchronous manifestation of HFRS epidemic activity in 2014, 2017.

**Figure 4 viruses-15-01537-f004:**
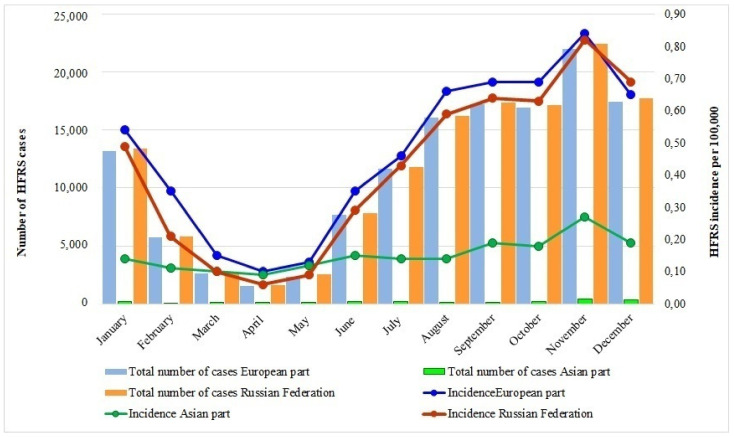
Monthly HFRS in 2000–2018.

**Table 1 viruses-15-01537-t001:** HFRS morbidity in Russia (2000–2022).

Area of Russia	No. of Regions with HFRS Cases	No. of HFRS Cases	Annual Average Morbidity Rate per 10^5^ Popul.	Children under 14 Years Old		Case Fatality	
				No. of HFRS Cases	%	No. of Cases	%
Total for Russia	69 out of 85	164,582	4.9	4030	2.4	668	0.4
European part	53 out of 60	162,044	9.7	3957	2.4	572	0.4
Asian part	16 out of 25	2538	0.6	73	2.9	96	3.8
Far-East	6 out of 8	2229	1.4	67	3.0	89	4.0
Western Siberia	6 out of 11	300	0.4	6	2.0	5	1.7
Eastern Siberia	4 out of 6	9	0.07	0	-	2	-

**Table 2 viruses-15-01537-t002:** Orthohantaviruses isolated in Russia.

Orthohantaviruses	Number of Strains	Source of Isolation
Puumala	12	Blood of HFRS patients; sectional organs of HFRS patients: kidney, lung, and lymph nodes; mammals: bank vole
Hantaan	30	Blood of HFRS patients; sectional organs of HFRS patients: kidney, lung, spleen, liver, and brain; mammals: striped field mouse, Korean field mouse, ruddy vole, red-backed vole, and yellow-throated bunting
Seoul	3	Mammals: *Rattus norvegicus*
Kurkino	7	Blood of HFRS patients; mammals: striped field mouse
Sochi	6	Blood of HFRS patients; mammals: Black Sea field mouse
Tula	8	Mammals: common vole
Khabarovsk	10	Mammals: reed vole, and red-backed vole
Total	76	Blood of HFRS patients; sectional organs of HFRS patients; 9 species of small mammals

## Data Availability

Data available in a publicly accessible repository that does not issue DOIs. Publicly available datasets were analyzed in this study. This data can be found using Federal Service for Surveillance on Consumer Rights Protection and Human Wellbeing. Available online: https://www.rospotrebnadzor.ru/activities/statistical-materials/ (accessed on 25 May 2023).

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
