# Peer review of "Cases of Hemorrhagic Fever with Renal Syndrome in Russia during 2000–2022"

_viruses, 2023, doi:10.3390/v15071537_

Round 1

Reviewer 1 Report

viruses-2445129: HEMORRHAGIC FEVER WITH RENAL SYNDROME IN RUSSIA, 2000-2022.

Evgeniy Tkachenko and collaborators performed a review synthesizing a large amount of information related to the HFRS in Russia recovered for more than 20 years. The article is a contribution to the virology community particularly for understanding the geographic spread of the pathogen, and providing valuable epidemiological information between the HFRS forms.

Commentaries

My main concern is that the authors should follow the hantavirus classification provided by Laenen et. al, viruses 2019. This include the specific nomenclature of the viruses mentioned in the text, such as Amur virus, Kurkino virus and Sochi virus. If the authors pursue to use those names, an explanation about that should be added in the text.

Other commenataries.

L 55-56: The number of reported hantaviral infection cases are increasing in many countries. The new hantavirus strains have been successfully identified worldwide. These sentences should have a citation.

L66. “10^5 popul.” It should be spell out the first time is written in the text.

L84: What do they mean with “unique geo-botanical conditions”? Please explain.

L139-140. The sentence “The difference may be explained primarily  by the fact that in women HFRS is frequently misdiagnosed as pyelonephritis and other diseases.” needs a citation.

L196-197: The sentence “Cell culture isolates of SOCV have been generated from Syl-196 vaemus ponticus and HFRS patient with fatal outcome# needs a citation.

L198-199: … 7 administrative Krasnodar area  regions? Please clarify “area regions”.

L200-201: Please clarify what do they mean with “representative of Dobrava-Belgrade  hantavirus”.

L205: “the age average” should be “average age”.

Figure 4. A paragraph explaining the highest incidence of HFRS cases during autumn and winter should be added to text.

Minor editing of English language is required

Author Response

First of all, we would like to thank the reviewer for their constructive comments. The comments were very helpful, and we took them into account when correcting our manuscript.

Commentary: “My main concern is that the authors should follow the hantavirus classification provided by Laenen et. al, viruses 2019. This include the specific nomenclature of the viruses mentioned in the text, such as Amur virus, Kurkino virus and Sochi virus. If the authors pursue to use those names, an explanation about that should be added in the text”.

Answer: Corrected (see manuscript abstract L 22-29)

Commentary: L 55-56: The number of reported hantaviral infection cases are increasing in many countries. The new hantavirus strains have been successfully identified worldwide. These sentences should have a citation.

Answer: Citations added (see L 60 and References #16-19)

Commentary: L66: “10^5 popul.” It should be spell out the first time is written in the text.

Answer: Value “105 popul.” was deciphered for the first time in the manuscript abstract L15.

Question: L84: What do they mean with “unique geo-botanical conditions”? Please explain.

Answer: The concept of "Unique geo-botanical conditions" means an excellent fodder potential for the bank voles - lime-forests.

Commentary: L139-140: The sentence “The difference may be explained primarily by the fact that in women HFRS is frequently misdiagnosed as pyelonephritis and other diseases.” needs a citation.

Answer: Citation added (see L 1540 and Reference #29)

Commentary: L196-197: The sentence “Cell culture isolates of SOCV have been generated from Sylvaemus ponticus and HFRS patient with fatal outcome needs a citation.

Answer: Citations added (see L 212 and References #38,47)

Question: L198-199: … 7 administrative Krasnodar area regions? Please clarify “area regions”.

Answer: Corrected (see the text L 213-214). For the period from 2000 to 2017, a total of 7 of Krasnodar region’s 37 districts reported 70 HFRS cases, including 38 HFRS cases in Sochi district.

Question: L200-201: Please clarify what do they mean with “representative of Dobrava-Belgrade hantavirus”.

Answer: Corrected (see the text L 218-219). At the present state of knowledge, SOCV seems to be the most dangerous in comparison with other 3 viruses (DOBV, KURV, SAAV) of Dobrava-Belgrade orthohantavirus species.

Commentary: L205: “the age average” should be “average age”.

Answer: Corrected (see the text L 225).

Commentary: Figure 4. A paragraph explaining the highest incidence of HFRS cases during autumn and winter should be added to text.

Answer: Text and citation added (see the text L 127-130 and Reference #23).

A significant decrease in autumn and winter seasons of feed products in the natural habitats of rodents - carriers of hantaviruses forces them to migrate to people's homes, which leads to an increase in their contacts with humans and an increase in the incidence of HFRS.

Comments on the Quality of English Language: Minor editing of English language is required

Answer: Corrected

Reviewer 2 Report

This review describes and summarizes the epidemiology of HFRS in Russia during the last two decades, contributing to a better knowledge of the incidence, fatality rate, and geographic distribution of HFRS throughout the country. In the Old and New World, outbreaks of hantavirus infections have been mainly associated with changes in rodent population density, which can vary significantly over time (seasonally or annually, depending on external factors such as interspecific competition, climate changes, and predatory anthropic activities). An important issue that could be mentioned is the prevalence of hantavirus in rodent hosts that are essential for understanding hantavirus dynamic transmission with respect to the environmental (climate conditions and landscape composition) and ecological (virus-reservoir interactions) contexts and with respect to human risk behavior that differ between the various regions in which illness is reported. The authors could reinforce their data with a geographic landscape composition map of the country and rodent hosts distribution, as a suggestion.  

I have outlined some specific comments below:

  1. Title: Substitute “Curent” to “Current”
  2. Line 46 and 172: Apodemus agrarius coreae (italicized)
  3. Line 69: the sentence “However, different geographical regions are distinguished by the morbidity rates due to HFRS very considerably” should be clarified or rewritten. It lacks information about the main factors that could be responsible for the differences in incidence rates (i.e., landscape structure and use, rodent reservoir geographic distribution, HFRS local surveillance).
  4. Table 1. I suggest excluding the line Total for Asia or include another column to include Asian part regions.
  5. Figure 1. “…in Russia from, (,) or (2000-2022). Scale bar should be included.
  6. Line 192/193/195: Apodemus ponticus (italicized)
  7. Figure 3. Please explain and correct “….activity in 2014, 2017…?” to “2000-2022.”
  8. Line 116: It is not necessary divide the text in subtopic 3.
  9. Line 123: Correct to “Hantaan”.
  10. Lines 137-144: Could the majority of cases occurring in residents of rural areas and males also be related to an occupational exposure?
  11. Lines 226-228: Please provide references to these data.
  12. I suggest including a figure illustrating the six hantaviruses and their principal hosts distribution areas to be more clear and informative about the circulation of the different species of hantavirus in Russia and associate it with the landscape composition.
  13. Lines 243-247: Considering that a hantavirus strain was isolated from some bird’s species, since it is a new and unusual information, more detail about the hantavirus specie or subtype could be provided to clarify whether the birds also have a role in their spread and create a potential health risk to humans.

 It was detected some minor incomprehensible words or sentences (i.e, "Curent", "...very considerably."...)

Author Response

Commentary: Title: Substitute “Curent” to “Current”

Answer: The title of the manuscript has been changed to: “Hemorrhagic fever with renal syndrome in Russia, 2000-2022”.

Commentary: Line 46 and 172: Apodemus agrarius coreae (italicized)

Answer: Corrected (see L 51)

Commentary: Line 69: the sentence “However, different geographical regions are distinguished by the morbidity rates due to HFRS very considerably” should be clarified or rewritten. It lacks information about the main factors that could be responsible for the differences in incidence rates (i.e., landscape structure and use, rodent reservoir geographic distribution, HFRS local surveillance).

Answer: Corrected (see the text L 76-78). However, different geographical regions are distinguished by the morbidity rates due to HFRS very considerably, which is mainly associated with the landscape zones (mixed and broad leaved forests, forest -steppe, steppe, semi-desert, desert and mountainous) rodent species - reservoir of hantaviruses, feed products for them, etc.

Commentary: Table 1. I suggest excluding the line Total for Asia or include another column to include Asian part regions.

Answer: Corrected (see Table 1)

Commentary: Figure 1. “…in Russia from, (,) or (2000-2022). Scale bar should be included.

Answer: Corrected (see Figure 1)

Commentary: Line 192/193/195: Apodemus ponticus (italicized)

Answer: Corrected (see L 26, 208, 235)

Commentary: Figure 3. Please explain and correct “….activity in 2014, 2017…?” to “2000-2022.”

Answer: The name of Figure 3 is correct so far as asynchronous manifestation of HFRS epidemic activity was revealed only in 2014 and 2017.

Commentary: Line 116: It is not necessary divide the text in subtopic 3.

Corrected (see the text)

Commentary: Line 123: Correct to “Hantaan”.

Answer: Corrected (see L 137).

Question: Lines 137-144: Could the majority of cases occurring in residents of rural areas and males also be related to an occupational exposure?

Answer: The main risk factors of HFRS infection in rural areas are short-time stay in the forest, gardening, farming, agricultural and hibernal cattle-breeding activities.

Commentary: Lines 226-228: Please provide references to these data.

Answer: Citation added (see L 250 and Reference #54)

Commentary: I suggest including a figure illustrating the six hantaviruses and their principal hosts distribution areas to be more clear and informative about the circulation of the different species of hantavirus in Russia and associate it with the landscape composition.

Answer:

In Section 2 (Lines 227-232) as well as in Section 3 “Epizootology” (Lines 251-258) presents detailed responses to the reviewer's proposal.

Commentary: Lines 243-247: Considering that a hantavirus strain was isolated from some bird’s species, since it is a new and unusual information, more detail about the hantavirus specie or subtype could be provided to clarify whether the birds also have a role in their spread and create a potential health risk to humans.

Answer: See Lines 258-269 and Table 2.

Comments on the Quality of English Language

It was detected some minor incomprehensible words or sentences (i.e, "Curent", "...very considerably."...)

Answer: Corrected

Reviewer 3 Report

The authors summarize interesting findings of the epidemiology of hantaviruses in Russia: Species, gender, age, seasonality, geographical distribution. However, in my opinion, the manuscript is too similar to the paper published in EID in 2019.

Author Response

Answer: The similarity of the articles lies only in the problem of HFRS in Russia. However, the amount and significance of the data presented in the 4 sections (epidemiology, epizootology, specific laboratory diagnosis, control and prevention) of the approximately 4,300 word manuscript submitted to the journal Viruses far exceeds those of the 700 word EID epidemiological summary.